# CEED-VLA: Consistency Vision-Language-Action Model with Early-Exit Decoding

## Abstract

The practical deployment of Vision-Language-Action (VLA) models is severely constrained by inference speed bottlenecks, particularly in high-frequency and dexterous manipulation tasks. While recent studies have explored Jacobi decoding as a more efficient alternative to traditional autoregressive decoding, its practical benefits are marginal due to the lengthy iterations. To address this problem, we introduce consistency distillation training to predict multiple correct action tokens in each iteration, thereby achieving acceleration. Besides, we design mixed-label supervision to mitigate error accumulation during distillation. While the above two methods bring obvious speedup, we identify that certain inefficient iterations remain a critical limitation. To tackle this, we propose an early-exit decoding strategy that moderately relaxes convergence conditions, which further improves average inference efficiency. Experimental results show that the proposed method achieves more than 4× inference acceleration across different base models while maintaining high task success rates in both simulated and real-world robot tasks. These experiments validate that our approach provides an efficient and general paradigm for accelerating multimodal decision-making in robotics.

## 1 Introduction

Recent advancements in Vision-Language Models (VLMs) (Liu et al., 2024c; Awadalla et al., 2023) have showcased impressive multimodal understanding capabilities, inspiring the development of Vision-Language-Action (VLA) models (Brohan et al., 2023; Zitkovich et al., 2023; Kim et al., 2024a; Driess et al., 2023; Black et al., 2024). These end-to-end architectures are trained on large-scale robotic datasets, integrating visual perception and language understanding to directly generate executable actions. Although VLA models generalize well across diverse tasks, their practical deployment is severely limited by inference speed bottlenecks, which hinder efficient execution and the handling of high-frequency, dexterous tasks. Therefore, our goal is to **significantly improve inference efficiency of VLAs while retaining the manipulation performance**.

To achieve this goal, recent works (Song et al., 2025; Kim et al., 2025) innovatively reframe autoregressive (AR) decoding as a system of nonlinear equations solved through the **Jacobi fixed-point iteration method** (Ortega & Rheinboldt, 2000). Specifically, the Jacobi decoding method begins by randomly initializing the $n$ action tokens in a sequence. Then, the action sequence along with the prompt is iteratively processed by the VLA to refine its predictions. Through $k$ successive updates, the $n$-token sequence converges to the fixed point, which is the same as the output produced by AR decoding under a greedy strategy. Because Jacobi decoding is parallel and allows the model to predict several correct tokens in each iteration, the number of iterations in Jacobi decoding can be fewer than the forward passes in AR decoding, i.e., $k \leq n$. This indicates that VLAs with Jacobi decoding can be faster than AR ones theoretically.

However, in practice, Jacobi decoding on standard VLAs (Song et al., 2025) delivers only limited acceleration over AR decoding, with recent studies reporting a modest $1.28\times$ speedup. This limitation arises because VLAs are trained in a strictly autoregressive manner, where models are only exposed to ground-truth prefixes during training. As a result, when preceding tokens are incorrect, which is a natural condition in Jacobi decoding, the model struggles to generate accurate predictions. Due to the lack of this capability, the model usually only correctly predicts the first token in each parallel iteration, resulting in slower convergence of the full token sequence, as shown in Figure 7.

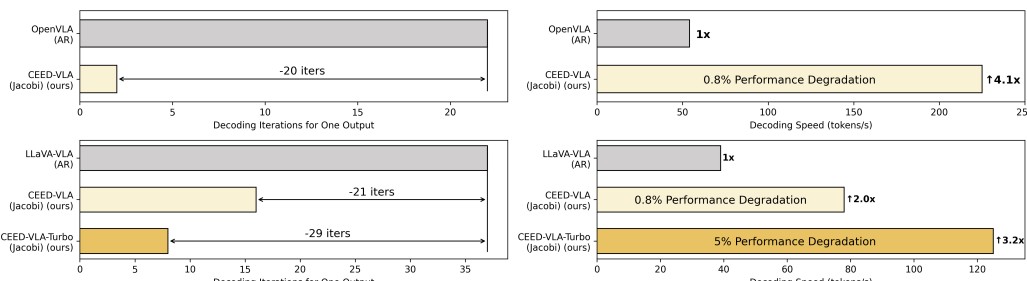

Figure 1: **Acceleration effect of our CEED-VLA on OpenVLA (Kim et al., 2024b) (the first row) and LLaVA-VLA (Song et al., 2025) (the second row)**. **Left**: Comparison of the number of iterations required for a complete output. CEED-VLA largely reduces the number of iterations, thus allowing faster decoding. **Right**: Comparison of the decoding speed. Our CEED-VLA separately realizes 3.6× and 2.0× speedup with negligible performance degradation on OpenVLA and LLaVA-VLA. In scenarios targeting more aggressive acceleration, CEED-VLA-Turbo delivers even fewer iterations and much more speedup while incurring only a slight degradation in performance.

To address this issue, we propose an acceleration strategy for VLAs based on **consistency distillation** process in Figure 2. This work adapts the consistency distillation technique on VLAs in order to enable the student model to predict several correct tokens in one iteration and shorten the decoding trajectory by learning from the decoding trajectory of the teacher model. We focus on two tasks to ensure an effective distillation process for VLAs: 1. Preserving the inherent autoregressive generation capability of VLAs while targeting consistency objectives during consistency training. 2. Eliminating unreliable supervision signals caused by errors accumulated during the collection of Jacobi trajectories. To mitigate these challenges, we propose a dual-supervision framework. First, we introduce an auxiliary AR loss to preserve the student model's native manipulation skills by aligning next-token distributions with those of the teacher model, thereby regularizing the distillation process. Second, we devise a mixed-label supervision mechanism that adaptively combines two sources of supervision. When the accuracy of the teacher model's action exceeds a learned threshold, its output is used as the supervisory signal. Otherwise, the target is replaced with the ground truth value. This adaptive strategy ensures robust student learning by selectively leveraging reliable teacher signals while mitigating the impact of performance degradation during distillation.

After consistency distillation, the student model demonstrates obvious acceleration effect, while the ideal acceleration is still bottlenecked by certain **redundant iterations**. Due to the strict convergence conditions of Jacobi decoding, these redundant iterations often require a large number of iterations to reach convergence, significantly reducing the average decoding speed. To address this issue, we propose an **early-exit decoding** strategy, which relaxes the strict convergence conditions of Jacobi decoding and thereby circumvents the impact of redundant iterations. Moreover, our analysis of the structural properties of the task and empirical observation shows that early-exit decoding has minimal effect on success rates, thus serving as a key accelerator. Our training approach closely resembles consistency models (Song et al., 2023), as both aim to accelerate inference by directly mapping intermediate equation states to the final solution. Thus, we term our model as **C**onsistency **V**ision-**L**anguage-**A**ction model with **E**arly-**E**xit **D**ecoding (**CEED-VLA**). A variant with a smaller exit point is termed as CEED-VLA-Turbo, featuring even faster inference and acceptable performance for tasks with relaxed precision requirements.

We also conduct experiments in two simulated environments, which show that CEED-VLA realizes **2-4.1×** acceleration with comparable success rates across different base models. Real-world experiments show that CEED-VLA realizes **4×** frequency in the practical robotic arm deployment with improved success rates on high-frequency dexterous tasks. We provide a straightforward visualization and detailed analyses, further revealing a key acceleration phenomenon. We also conduct extensive ablation studies to demonstrate the effectiveness of our key designs.

To summarize, our key technical contributions are as follows: 1) We propose CEED-VLA, a universal acceleration method for significant inference speedup while maintaining manipulation performance. 2) We conduct a consistency distillation process to unlock the model's capabilities of fast inference, and we further propose a mixed-label supervision in the autoregressive loss to preserve the model's manipulation performance. 3) We identify the redundant iterations as the bottleneck of Ja-

cobi decoding's speedup and propose the early-exit decoding to solve it, resulting in $4.1\times$ speedup, and more than $4.3\times$ frequency.

## 2 RELATED WORK

### 2.1 ACCELERATION FOR VISION-LANGUAGE-ACTION MODELS

Various acceleration strategies, including quantization (Lin et al., 2024b) and token pruning (Chen et al., 2024), have been effectively applied to LLMs, yet they often fail to meet the stringent real-time requirements of action generation. Efforts to enhance efficiency have led to architectural modifications in VLA models, such as DeeR-VLA (Yue et al., 2024), which dynamically adjusts inference depth, and QAIL (Park et al., 2024), which integrates quantization-aware training. Further innovations, like RoboMamba (Liu et al., 2024d) and TinyVLA (Wen et al., 2025), replace traditional attention mechanisms or focus on developing lightweight models from the ground up, frequently necessitating model re-training and additional data collection. Meanwhile, VLA-Cache (Xu et al., 2025) selectively caches static tokens and recomputes only dynamic or task-relevant ones. FAST (Pertsch et al., 2025) proposes a compression-based tokenization scheme based on the discrete cosine transform. MoLe-VLA (Zhang et al., 2025b) achieves this by selectively skipping transformer layers based on task-relevant cues. EfficientVLA (Yang et al., 2025) first designs a systematic method to eliminate multifaceted redundancies across the visual encoder, language model, and action head. PD-VLA (Song et al., 2025) framework reformulates autoregressive decoding as a nonlinear system and solves it using a parallel fixed-point iteration method, significantly improving decoding speed while maintaining model performance. OpenVLA-OFT (Kim et al., 2025) employs a similar parallel decoding method to speed up. By contrast, our CEED-VLA unlocks the acceleration potential of VLAs by fine-tuning them with a consistency distillation process. It also optimizes the decoding mechanisms, leading to more significant acceleration.

### 2.2 CONSISTENCY MODELS FOR MANIPULATION

In manipulation tasks, consistency policy (Prasad et al., 2024) serves as a key acceleration technique for diffusion policies, which adapts consistency models (Song et al., 2023) from image generation fields to robotics. It enables single-step inference in a student model while analyzing the impact of design choices like objectives, variance, and chain steps on consistency distillation. ManiCM (Lu et al., 2024) applies consistency constraints to the diffusion process to enable fast inference without compromising action quality. FlowPolicy (Zhang et al., 2025a) enables single-step generation by normalizing velocity field consistency, refining flow dynamics for efficient inference. SDM policy (Jia et al., 2024) integrates score and distribution matching through a dual teacher framework to improve the speed of inference and the quality of action. While these consistency policies directly map intermediate states of ordinary differential equations (ODEs) to their final solution, we train VLAs to map the intermediate points in the Jacobi trajectory to the fixed point. This paper takes the first step to explore consistency training techniques for efficient VLAs.

## 3 CEED-VLA

### 3.1 PRELIMINARY: JACOBI DECODING

Given a prompt $\boldsymbol{x}$, an autoregressive LLM $p$ typically generates responses in a next-token-prediction manner:

$$y_i = \arg\max_y p(y|\mathcal{Y}_{i-1}, \boldsymbol{x}) \text{ for } i = 1, \ldots, n, \tag{1}$$

where $\mathcal{Y}_{i-1}$ denotes $\{y_1, \ldots, y_{i-1}\}$, $n$ represents the number of tokens to predict. Thus, LLM executes $n$ forward passes to obtain $n$ tokens $\mathcal{Y}_n$, which makes it hard to efficiently output a lengthy token sequence.

Unlike AR decoding, Jacobi decoding (Santilli et al., 2023; Kou et al., 2024) can accelerate the inference by predicting several tokens in one forward. To directly predict several tokens in each forward, Equation (1) is reformulated as a system of nonlinear equations with respect to $y_i$:

$$\begin{cases} y_1^{(j+1)} &= \arg\max_y \ p(y \mid \boldsymbol{x}) \\ y_2^{(j+1)} &= \arg\max_y \ p(y \mid \mathcal{Y}_1^{(j)}, \boldsymbol{x}) \\ &\vdots \\ y_n^{(j+1)} &= \arg\max_y \ p(y \mid \mathcal{Y}_{n-1}^{(j)}, \boldsymbol{x}), \end{cases} \tag{2}$$

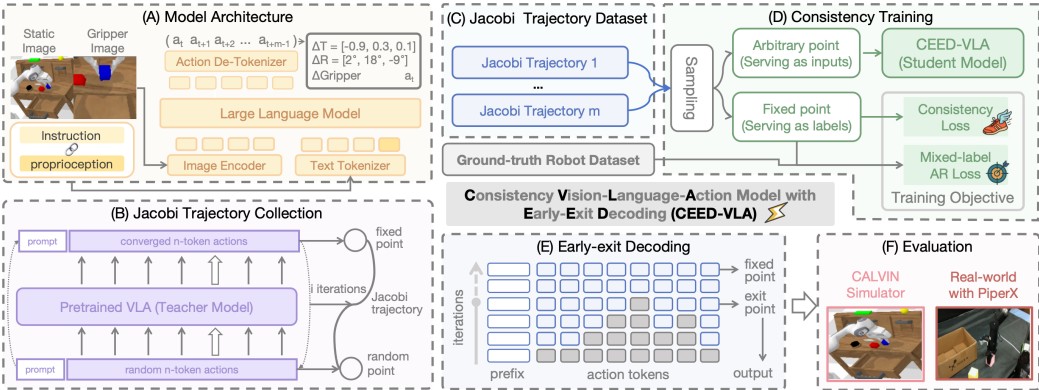

Figure 2: Overview of our proposed **CEED-VLA**. Given an autoregressive VLA model (*e.g.*, LLaVA-VLA) that is pretrained on an in-domain task (**A**), we first prompt it to perform Jacobi decoding to collect Jacobi trajectory dataset (**B, C**). Then we design an effective consistency training process with novel mixed-label supervision to train a student model (**D**). Finally, we propose early-exit decoding to further unlock inference speed (**E**). Experiments in simulators and the real world show significant acceleration with comparative success rates (**F**).

where $j$ denotes the $j$-iteration of the Jacobi trajectory. This enables updates of every token in each forward iteration until convergence. The nonlinear equation system can be solved in the Jacobi fix-point iteration method (Ortega & Rheinboldt, 2000). Concretely, Jacobi decoding first initializes a random token sequence $\mathcal{Y}^{(0)} = \{y_1^{(0)}, \ldots, y_n^{(0)}\}$. Then, both the prompt $x$ and the token sequence are fed into the LLM simultaneously. Then, the variables $y_i$ in $\mathcal{Y}$ are iteratively updated through Equation (2) until convergence. The convergence condition is $\mathcal{Y}^{(k)} = \mathcal{Y}^{(k-1)}$ at the step $k$. The $\mathcal{Y}^* := \mathcal{Y}^{(k)}$ is defined as the fixed point. Because Jacobi decoding allows to predict multiple correct tokens in the $n$-token sequence in parallel, it requires fewer iterations than AR decoding, thus improving the inference speed.

## 3.2 TEACHER MODEL

Vanilla VLAs $P_\gamma$ learn to predict actions $\hat{a}_{t+1}$ directly from the current observation $s_t$ and language instruction $l$ (See Figure 2). Some research (Song et al., 2025) replaces the AR decoding in these VLAs with Jacobi decoding to accelerate the inference, while the speedup is **limited** ($1.28\times$). The reason is that VLAs, which have been trained autoregressively, struggle to produce multiple correct tokens within a single Jacobi iteration: each newly generated token is conditioned on the previously generated ones. Therefore, when any preceding token is incorrect, the model is unlikely to generate the correct subsequent token.

## 3.3 STUDENT MODEL

To address the aforementioned issue and fully unlock the acceleration potential, we finetune VLAs to empower them to output multiple correct actions with wrong preceding tokens. This section first illustrates the data collection for tuning CEED-VLA and then details the training procedure. Finally, we propose an efficient decoding strategy to further accelerate inference.

For the target VLA $P_\gamma$, we let $Q_\theta(\cdot|s_t, l)$ denote the CEED-VLA with parameters $\theta$ initialized with those of $P_\gamma$. To capture the inherent consistency within Jacobi trajectories, we first collect them by prompting $P_\gamma$ to predict actions with Jacobi decoding on the robot dataset $\mathcal{C}$. We record the Jacobi trajectories during decoding to build the consistency distillation dataset $\mathcal{D}$. The dataset construction process is detailed in Algorithm 2.

### 3.3.1 CONSISTENCY TRAINING

The consistency training procedure optimizes two objectives: *(i)* guiding the model to predict multiple correct tokens simultaneously, and *(ii)* constraining CEED-VLA from drifting away from the

target VLA distribution to preserve manipulation skills. To achieve this goal, we meticulously designed two losses.

We first introduce our consistency loss. The ideal finetuned model consistently maps any point $\mathcal{Y}$ on the Jacobi trajectory $\mathcal{J}$ to the fixed point $\mathcal{Y}^*$. We design the consistency loss based on this notion. Given a prefix $x$ consisting of both instructions and visual inputs, and its Jacobi trajectory $\mathcal{J}$, let $\mathcal{Y}$ be a randomly sampled intermediate state and $\mathcal{Y}^*$ the corresponding fixed point. We train CEED-VLA to directly predict $\mathcal{Y}^*$ from $\mathcal{Y}$ by minimizing the following loss:

$$\mathcal{L}_{\mathrm{C}} = \mathbb{E}_{(s_t,l,\mathcal{J})\sim\mathcal{D},\, \mathcal{Y}\sim\mathcal{J}} \left[ \sum_{i=1}^{n} Q_{\theta^-}(\cdot|\mathcal{Y}_i^*, s_t, l) \log \frac{Q_{\theta^-}(\cdot|\mathcal{Y}_i^*, s_t, l)}{Q_\theta(\cdot|\mathcal{Y}_i, s_t, l)} \right], \qquad (3)$$

where $\theta^- = \mathrm{stopgrad}(\theta)$.

Now we introduce our mixed-label AR supervision. To avoid deviating from the distribution of the teacher model, we inherit the AR loss function $\mathcal{L}_{\mathrm{AR}}$ used for training the teacher model:

$$\mathcal{L}_{\mathrm{AR}} = \mathbb{E}_{(s_t,l,\mathcal{Y}^*)\sim\mathcal{D}} \Big[ -\sum_{i=1}^{N} \log Q_\theta(\mathcal{Y}_i^*|\mathcal{Y}_{<i}^*, s_t, l) \Big] \qquad (4)$$

To address potential performance degradation stemming from teacher model inaccuracies in action prediction, we implement a mixed-label supervision strategy. Specifically, we compute the $L_1$ distance to quantify the distributional divergence between the generated Jacobi trajectory dataset and the corresponding ground-truth action data (See Figure 6). For high-deviation samples, we replace the AR loss labels of those samples with the corresponding ground-truth values. We define a correctness threshold $\delta_{\max}$, and any data in the dataset $\mathcal{D}$ with an $L_1$ distance exceeding this threshold is considered an outlier. This strategy effectively mitigates error propagation during consistency distillation, maintaining generation quality substantially. Consequently, the total loss for training our CEED-VLA is:

$$\mathcal{L}(\theta) = \mathcal{L}_{\mathrm{C}} + w\mathcal{L}_{\mathrm{AR}}, \qquad (5)$$

where $\omega$ represents a weighting coefficient. The training procedure is detailed in Algorithm 1.

---

**Algorithm 1** Mixed-label training

1: **Input:** Original robot dataset $\mathcal{C}$, Jacobi trajectory dataset $\mathcal{D}$, weight $\omega$, CEED-VLA $Q_\theta(\cdot \mid s_t, l)$, correctness threshold $\delta_{\max}$
2: **repeat**
3:     Sample observation $s_t$ and language instruction $l$, Jacobi trajectory $\mathcal{J}$ from $\mathcal{D}$, teacher model's output $\mathcal{Y}^*$ from $\mathcal{J}$, ground-truth label GT from $\mathcal{C}$
4:     **if** $\mathrm{L1}(\mathcal{Y}^*, \mathrm{GT}) < \delta_{\max}$ **then**
5:         Compute $\mathcal{L}_{\mathrm{AR}}$ using Equation (4)
6:     **else**
7:         Reformulate Equation (4) as $\mathcal{L}_{\mathrm{AR}} = \mathbb{E}_{(s_t,l,\mathrm{GT})}$
8:     **end if**
9:     Randomly sample $\mathcal{Y}$ from $\mathcal{J}$
10:     Compute $\mathcal{L}_{\mathrm{C}}$ using Equation (3)
11:     update parameters $\theta$ using $\mathcal{L}$ (Equation (5))
12: **until** convergence

---

### 3.3.2 INFERENCE

We first introduce the existence of redundant iterations. Although the student model CEED-VLA performs inference via Jacobi decoding after consistent distillation, our empirical observations reveal that the acceleration performance of CEED-VLA is bottlenecked by a few iterations that are even slower than the minimum speed of AR decoding (see Table 1). This phenomenon arises from the **strict convergence conditions** of the fixed point. It requires exact convergence between successive iterations, *i.e.*, $\mathcal{Y}^{(k)} = \mathcal{Y}^{(k-1)}$, which takes numerous iterations (*over 30 iterations*) to reach. It is further observed that token updates in the final steps of redundant iterations are limited, contributing little to the final action decisions. These redundant iterations significantly hinder the speedup of inference and may introduce delays or discontinuities in high-frequency dexterous tasks, ultimately compromising task performance.

Table 1: Comparison of inference speeds among AR decoding, Jacobi decoding, and early-exit decoding on our CEED-VLA.

| Decoding Method | Avg. Speed | Min. Speed | Max. Speed |
|---|---|---|---|
| AR | 39.64 | 30.87 | 49.10 |
| Jacobi | 57.57 ↑ | 19.50 ↓ | 82.64 ↑ |
| Early-exit | 79.24 ↑ | 62.07 ↑ | 93.53 ↑ |

Table 2: Comparison with various base models in terms of inference speed, success rate or average length, and execution frequency. Following (Song et al., 2025; Kim et al., 2024b), experiments based on LLaVA-VLA are conducted on the CALVIN (Mees et al., 2021) ABC→D dataset using a NVIDIA 4090 GPU, while those based on OpenVLA are evaluated on the LIBERO (Liu et al., 2024a) Long dataset using a NVIDIA H100 GPU. Compared methods are detailed in Appendix H.

| Acceleration Techniques | Action Chunking | Decoding Method | Splits | Speed (Tokens/s) | Avg. Len. | SR (%) | Freq. (Hz) |
|---|---|---|---|---|---|---|---|
| Base model: **LLaVA-VLA** | | | Benchmark: **CALVIN** | | | | |
| N/A | - | AR | ABC→D | 39.6 (×1) | 2.01 | – | 2.23 (×1) |
| N/A | 5 | AR | ABC→D | 39.6 (×1) | **3.70** | – | 4.33 (×1.9) |
| FastV (Chen et al., 2024) | 5 | AR | ABC→D | 28.7 (×0.7) | 2.54 | – | 1.87 (×0.8) |
| SparseVLM (Zhang et al., 2025c) | 5 | AR | ABC→D | 32.4 (×0.8) | 2.83 | – | 2.01 (×0.9) |
| PD-VLA (Song et al., 2025) | 5 | Jacobi | ABC→D | 52.8 (×1.3) | 3.69 | – | 5.43 (×2.4) |
| CEED-VLA (Ours) | 5 | Jacobi | ABC→D | **79.2** (×2.0) | 3.67 | – | **7.27** (×3.3) |
| Base model: **OpenVLA** | | | Benchmark: **LIBERO** | | | | |
| N/A | - | AR | Long | 54.4 (×1) | – | 53.2 | 5.95 (×1) |
| N/A | 3 | AR | Long | 54.4 (×1) | – | 60.4 | 7.08 (×1.2) |
| PD-VLA (Song et al., 2025) | 3 | Jacobi | Long | 85.0 (×1.6) | – | **62.4** | 10.79 (×2.4) |
| CEED-VLA (Ours) | 3 | Jacobi | Long | **225.0** (×4.1) | – | 62.2 | **25.60** (×4.3) |

Then we introduce our proposed early-exit decoding technique. To address the aforementioned limitation, we investigate whether relaxed convergence conditions or fewer iterations are feasible, particularly in terms of their impact on overall performance. We base our investigation on two aspects: 1. Theoretical structural properties. Unlike popular VQA tasks (*e.g.* multiple-choice), where each inference has a direct correctness, the success of a robotic manipulation task is decided on a sequence of actions. Within it, an **overlooked** yet critical insight is that task success is primarily determined by actions executed at a small number of key states, *e.g.*, releasing the gripper at the target location. In contrast, the majority of states along a trajectory do not demand optimal actions, or the set of "good-enough" actions is relatively large (Kumar et al., 2022; Zhao et al., 2025). Therefore, slight degradation in the quality of certain actions does not adversely affect task outcomes. 2. Empirical observation. We further observe that token updates in the final steps of redundant iterations are limited and contribute little to the final action decisions, further highlighting their inefficiency. This suggests that relaxing convergence and reducing iterations introduce only minor degradation in action quality. Based on the above analysis and empirical observations, we conclude that relaxing convergence conditions and reducing iterations are feasible without compromising performance.

To this end, we propose **early-exit decoding** to relax the condition and significantly reduce the number of iterations. Specifically, we introduce the exit point $\sigma$, at which the model prematurely halts the iterative process and directly emits the intermediate output at the $\sigma$-th iteration, bypassing further decoding steps. This approach mitigates redundant iterations, significantly enhancing both the minimum and average decoding speeds while maintaining performance. We further conduct ablation studies to investigate the optimal value of the exit point, see Section 4.2. When the exit point is set to a small value, the model achieves even higher speedup with an acceptable performance drop, termed as CEED-VLA-Turbo. In addition, since the prefix tokens remain unchanged across iterations, we cache and reuse their attention key–value states, reducing redundant computation.

# 4 EXPERIMENTS

## 4.1 SIMULATED EXPERIMENTS

**Benchmarks and Metrics.** To evaluate the success rates and inference speedup of our CEED-VLA, we carefully selected two widely used simulation benchmarks in the robot learning field and VLA tasks for comprehensive experiments. The CALVIN benchmark (Mees et al., 2021) includes 34 tasks across four environments (A, B, C, and D). Following the classic ABC→D setup, we run 500 rollouts per model, where each rollout involves a sequence of 5 consecutive sub-tasks. We

Table 3: Ablation of different techniques used in our CEED-VLA applied to the base model Open-VLA of (Kim et al., 2024b) on LIBERO LONG (Liu et al., 2024a) tasks.

| Jacobi Decoding | Consistency Training | Mixed-Label Supervision | Early-exit Decoding | Speedup | Success Rate(%) |
|---|---|---|---|---|---|
| ✓ | ✓ | ✓ | ✓ | ×4.1 | 62.2 |
| ✓ | ✓ | ✓ | × | ×2.5 | 61.2 |
| ✓ | ✓ | × | × | ×2.3 | 54.2 |
| ✓ | × | × | × | ×1.6 | 62.3 |
| × | × | × | × | ×1.0 | 60.4 |

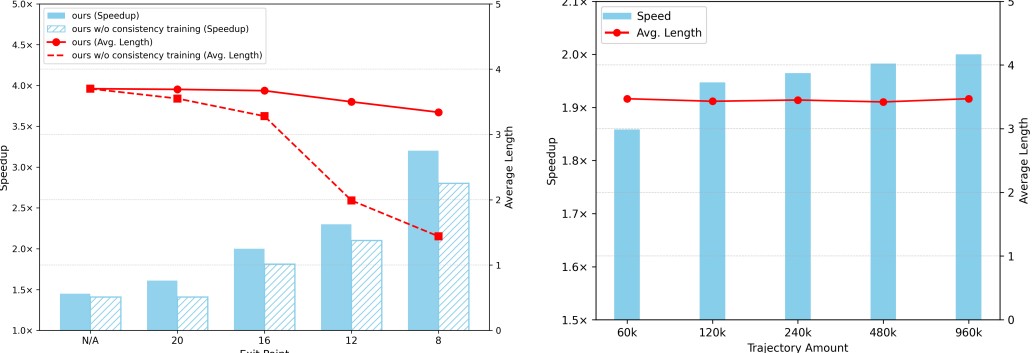

Figure 3: In-depth ablation experiments on exit point and consistency training (left), and on different amounts of training data (right). This figure reports speedup and average success length using LLaVA-VLA (Song et al., 2025) as the base model on the CALVIN (Mees et al., 2021) benchmark. report the average length of successful sub-task completions of all rollouts, denoted as avg. len., with a maximum value of 5. LIBERO-Long (Liu et al., 2024a) consists of 10 long-horizon and multi-step manipulation tasks with diverse objects and skills, emphasizing temporal reasoning, goal consistency and delayed rewards. Each method is evaluated over 50 rollouts per task, for a total of 500 rollouts, and we report the overall average success rate.

**Base Models.** In this section, we consider fine-tuning diverse base models for a comprehensive validation of our CEED-VLA. LLaVA-VLA is a vanilla VLA model based on LLaVA (Liu et al., 2024c), with further details provided in our supplementary material. We also include OpenVLA (Kim et al., 2024a), the most popular open-source VLA model, for general evaluation. We further reproduce an OpenVLA with action chunking, which enables more stable actions.

**Acceleration Results on CALVIN.** We compare our method with naïve base models, advanced acceleration methods for VLMs (FastV (Chen et al., 2024), SparseVLM (Zhang et al., 2025c)), and VLAs in parallel decoding (PD-VLA (Song et al., 2025) ). As shown in Table 2, the key findings on CALVIN are as follows: 1. Only Jacobi decoding without consistency training leads to a modest improvement in VLA decoding speed. 2. With mixed-label consistency training and early-exit decoding, CEED-VLA is able to predict more fixed points, resulting in a 2.0× speedup, 3.3× frequencies while the average length only reduces 0.03.

**Acceleration Results on LIBERO.** Experiments on the LIBERO-Long benchmark further validate our conclusions. Our proposed CEED-VLA realizes 4.1× speed up and 4.3× frequency, which is more evident than CALVIN. This is because the tasks on LIBERO are relatively easy and short-horizon, thus requiring fewer iterations in Jacobi decoding. Meanwhile, the integration with action chunking improves 6.8% success rates because of stronger planning abilities and smoother actions.

**Discussion and Visualization.** Moreover, we provide a detailed discussion of the acceleration gains achieved by different baselines and the underlying principles in Appendix E and a visualization of different decoding methods in Appendix F.

## 4.2 ABLATION STUDY

In this section, we first evaluate the significance of several key designs in our method with base model OpenVLA on LIBERO. As illustrated in Table 3, consistency training and early-exit de-

coding substantially accelerate the model, whereas mixed-label supervision plays a crucial role in maintaining action accuracy and success rate. On LLaVA-VLA, we investigate the choice of exit points and data amounts in Figure 3, consistency training in Table 1, supervising labels in Table 4, and losses in Table 5.

**Early-exit Decoding.** Early-exit decoding serves as a key accelerator, balancing inference speed and success rates. In Figure 3, we conduct inference on different values $\sigma$ of exit points. AR decoding requires 37 iterations to complete one set of action prediction because it needs to generate 5 groups of 7-dimensional action tokens along with start and end symbols. We observe that introducing early-exit decoding and setting $\sigma$ to 16, *i.e.*, a maximum of 16 iterations, leads to a noticeable speedup with negligible performance drop. This suggests that enabling early exit at a few steps within a task does not affect overall success. Reducing the $\sigma$ further to 8 results in a slight performance degradation but brings even greater acceleration. We term this version as CEED-VLA-Turbo.

**Consistency Training.** To underscore its necessity, we further conducted an ablation study on the model without consistency training under various exit point settings. Figure 3 shows that the average trajectory length decreases significantly as the exit point occurs earlier. Specifically, when the exit point is set to 12 and 8, the average lengths drop to 1.99 and 1.44, respectively, whereas our model with consistency training maintains an average length above 3.34. This indicates that consistency training plays a central role in enabling parallel decoding within CEED-VLA, allowing the model to produce accurate actions with minimal forward passes.

**Mixed-label Supervision.** We investigate the impact of label choices for supervising the AR loss. Table 4 shows that our mixed-label supervision best balances between speed-up and performance, because it prevents the student model from drifting from the ground truth. Using only the teacher output as the label causes error accumulation, resulting in shorter average episode lengths. Conversely, using only the ground truth creates conflicting supervision between the AR and consistency losses, yielding the worst performance.

Table 4: Ablation study of mixed-label supervision for the AR loss component in our CEED-VLA.

| Label | Speedup | Avg. Len. |
|---|---|---|
| Mixed (ours) | **2.00×** | **3.67** |
| Teacher Model | 1.83× | 3.48 |
| GT | 1.67× | 3.20 |

**Loss Design.** The consistency loss guides the model to learn the convergence behavior of the Jacobian trajectory toward a fixed point, thereby reducing iterations and accelerating the overall decoding process. The autoregressive (AR) loss encourages the model to learn correct actions, preventing it from deviating from the distribution of teacher model and ensuring performance. The trade-off between speed and accuracy is influenced by the relative weighting of the two losses. We experimented with weight ratios of 1 and 10, respectively. As shown in Table 5, increasing the emphasis on the AR loss does indeed improve accuracy, albeit at the cost of speedup.

Table 5: Comparison of inference speed and average length between different ratios of losses to evaluate the trade-off.

| Loss | Speedup | Avg. Len. |
|---|---|---|
| $\mathcal{L}_C + \mathcal{L}_{AR}$ (ours) | **2.00×** | 3.67 |
| $\mathcal{L}_C + 10 * \mathcal{L}_{AR}$ | 1.79× | **3.74** |

**Data Size.** CEED-VLA learns to predict multiple tokens in each step by leveraging pre-collected Jacobian trajectories. As shown in Figure 3, 60K trajectories are sufficient to achieve significant acceleration, demonstrating strong data efficiency because training for one epoch on this dataset takes only 10 hours. When the number of trajectories exceeds 120k, the performance gain plateaus and becomes marginal. Another observation is that increasing the data volume has a limited impact on the average episode length.

### 4.3 REAL-WORLD EXPERIMENTS

**Real-world Setup**. Our real-world experiments were performed on a dual-arm AgileX PiPer robotic system (See Figure 5). During the data collection phase, one arm is teleoperated by a human to provide demonstrations (designated as the lead arm), while the other arm (designated as the follower arm) remains passive. During evaluation, only the follower arm is controlled by VLAs to perform the tasks. To provide visual observations, we employ a RealSense L515 depth camera as third-view input and an ORBBEC Dabai depth camera as first-view input.

Table 6: Success rates in real-world experiments of our CEED-VLA, compared with the base model LLaVA-VLA and baseline method PD-VLA proposed in (Song et al., 2025). We evaluate all methods on 4 tasks, with 20 rollouts per task (*i.e.*, 80 rollouts in total) and report success rates (%).

| Method | Push button (basic) | Lift block (basic) | Pour water (dexterous) | Fold towel (dexterous) | Average | Frequency (Hz) |
|---|---|---|---|---|---|---|
| LLaVA-VLA | 65 | 40 | 15 | 5 | 33.25 | 3.3 |
| PD-VLA | 85 | 65 | 45 | 35 | 57.50 | 6.0 *(1.8×)* |
| CEED-VLA (ours) | **85** | **70** | **80** | **75** | **77.50** | **13.0** *(3.9×)* |

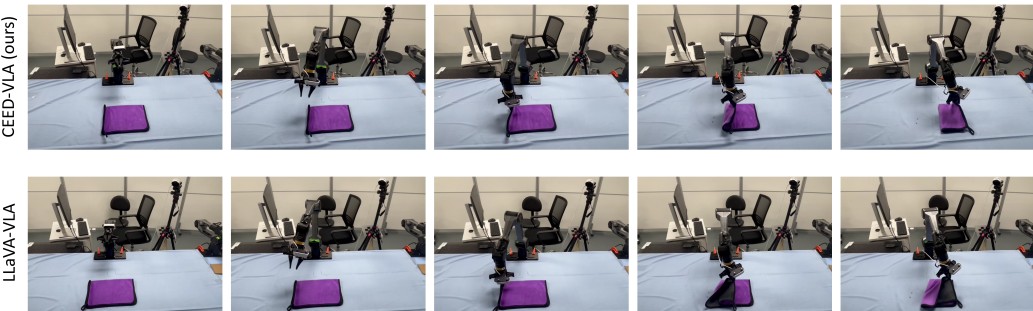

Figure 4: **Visualization of the dexterous task in the real-world experiment**. We take the towel folding for example. The first row illustrates a successful case of our CEED-VLA, while the second row presents a failure case of LLaVA-VLA under the same instruction and scene. Besides, the duration of the bottom figure is approximately 4× that of the top figure.

**Tasks and Datasets.** Our dataset covers two levels of manipulation difficulty, categorized into basic tasks and dexterous tasks, as described below. Basic tasks are short-horizon, atomic interactions such as button pushing and block lifting. These tasks focus on simple object interactions under basic planning and are collected at a frequency of 10 Hz. Dexterous tasks demand high-frequency, continuous control, and fine-grained manipulation skills. These tasks include towel folding and water pouring, which are collected at 30 Hz.

**Results.** Table 6 presents the real-world results. We observe that LLaVA-VLA achieves decent success rates on basic tasks, but struggles to learn effective policies with high-frequency robot data. Its discontinuous actions often lead to task failures. PD-VLA improves action continuity and execution frequency by integrating parallel decoding with action chunking, leading to higher success rates on both basic and dexterous tasks. Our CEED-VLA significantly boosts inference speed and control frequency, enabling the model to learn and execute high-frequency actions. As a result, it substantially improves success rates on dexterous tasks, exceeding 70%.

**Visualization.** To intuitively analyze the performance gain of CEED-VLA on dexterous tasks brought by accelerated inference, we visualize two representative cases in Figure 4. In the *fold towel* task, CEED-VLA successfully grasps the towel and completes the folding process with smooth actions. In contrast, although LLaVA-VLA is also capable of picking up the towel, its unstable motion causes one side of the towel to slip during the folding process, ultimately resulting in failure. The high-frequency inference of CEED-VLA not only shortened the task completion time, but also mitigated action latency, which induced pauses and jitters in lightweight robotic arms (*e.g.*, the PiPer arm shown in the figure). This issue is usually overlooked on industrial robotic arms.

## 5 CONCLUSIONS

In this paper, we propose CEED-VLA, a universal acceleration method for significant inference speedup while maintaining manipulation performance. Extensive experiments in several simulation and real-world environments demonstrate that our CEED-VLA better manages high-frequency dexterous tasks. We provide more discussion about future work in the Appendix G.

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

## A  LLM Usage Statement

We used large language models only as general-purpose writing aids for light proofreading and language refinement, including grammar, phrasing, and minor style adjustments. The research ideas, problem formulation, methodology, experiments, analyses, code, and substantive writing were conceived and authored by us. No LLM was used to generate new content beyond local edits, nor to design experiments, analyze data, write code, or draft sections. All text was reviewed and verified by the authors, who take full responsibility for the paper, and the LLM should not be regarded as a contributor.

## B  Real-world Setup

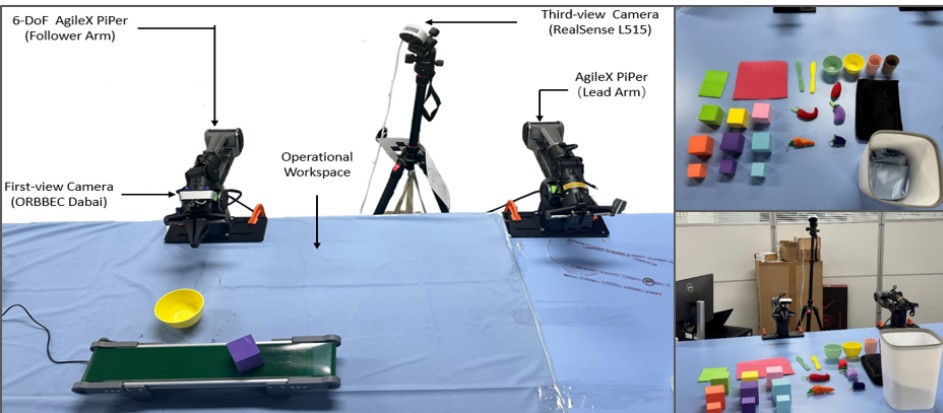

Figure 5: **The real-world robotic system for experiments**. The system consists of two AgileX PiPer 6-DoF robotic arms, an ORBBEC Dabai depth camera, and a RealSense L515 depth camera.

## C  Jacobi Trajectory Collection

---

**Algorithm 2** Jacobi trajectory collection

---

1: **Input:** Dataset $\mathcal{C}$, teacher model $P_\gamma$
2: **repeat**
3:     Sample observation $s_t$ and instruction $l$ from $\mathcal{C}$
4:     Initialize $\mathcal{Y}^{\{0\}}$ randomly
5:     **repeat**
6:         $\mathcal{Y}^{\{i\}} \leftarrow P_\gamma(\mathcal{Y}^{\{i-1\}}, s_t, l)$
7:         Add $\mathcal{Y}^{\{i\}}$ to $\mathcal{J}$
8:     **until** $\mathcal{Y}^{\{i\}} = \mathcal{Y}^{\{i-1\}}$
9:     Get complete $\mathcal{J} = \{\mathcal{Y}^{\{0\}}, \ldots, \mathcal{Y}^*\}$
10:    Append $(s_t, l, \mathcal{J})$ to $\mathcal{D}$
11: **until** all samples in $\mathcal{C}$ are processed

---

## D  NECESSITY OF MIXED-LABEL TRAINING

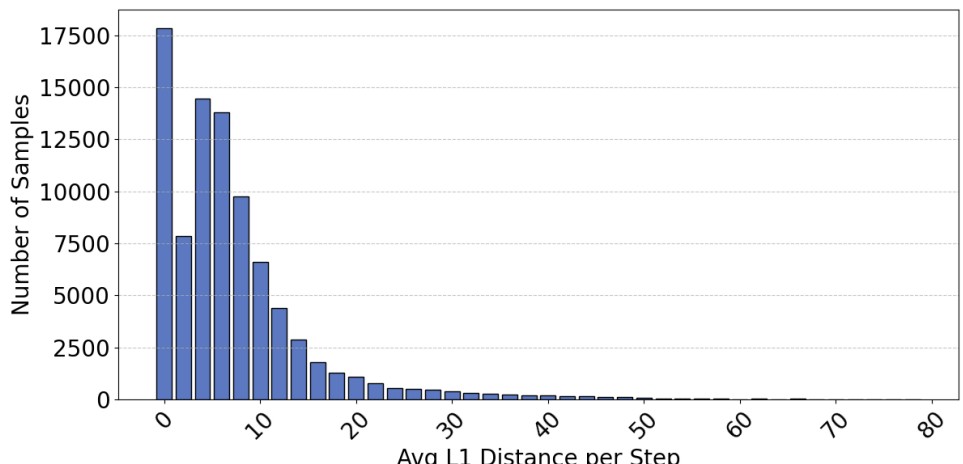

Figure 6: $L_1$ distance between the generated Jacobi trajectory dataset and the ground-truth data.

We observe that there is a discrepancy between the output of the teacher model and the ground truth. To ensure that the distribution learned by the student model does not diverge from the ground truth, we replace labels with large L1 distances with the ground truth as supervision for $\mathcal{L}_{AR}$.

## E  MORE ANALYSIS OF EXPERIMENTS

Table 7: Analysis of different decoding horizons on the CALVIN benchmark, based on the LLaVA-VLA model.

| Decoding Horizon | Average length | Avg. Speed (Token/s) | Frequency (Hz) |
|---|---|---|---|
| 7 | 3.24 | 41.48 | 3.60 |
| 16 | 3.19 | 48.74 | 3.25 |
| 37 | **3.69** | **52.84** | **5.43** |

Table 8: Profiling results for fixed tokens and speedups with LLaVA-VLA, PD-VLA, and our method.

| Model | Fixed Tokens | Speedup |
|---|---|---|
| LLaVA-VLA | 0 | 1× |
| PD-VLA | 8.75 | 1.33× |
| CEED-VLA (ours) | 13.5 | **2.00×** |

**Discussion: Why does the acceleration effect of our CEED-VLA on OpenVLA is more significant than that on LLaVA-VLA?** OpenVLA achieves more than 4× acceleration, in contrast to LLaVA-VLA's 2× speedup. The reason may be that this difference stems from the nature of the evaluation tasks: CALVIN features more complex, long-horizon scenarios composed of five subtasks, which demand more careful planning during execution, thereby increasing the number of required iterative steps.

**Underlying Acceleration Phenomena.** In Figure 7, we observe that our CEED-VLA is capable of correctly predicting certain action tokens (underlined blue numbers) in one iteration, even if preceding tokens are incorrect. We refer to such tokens as *fixed tokens*. The presence of fixed tokens significantly enhances the model's ability to predict multiple tokens within each iteration. Table 8 presents the correlation between the number of fixed tokens and the resulting acceleration gains.

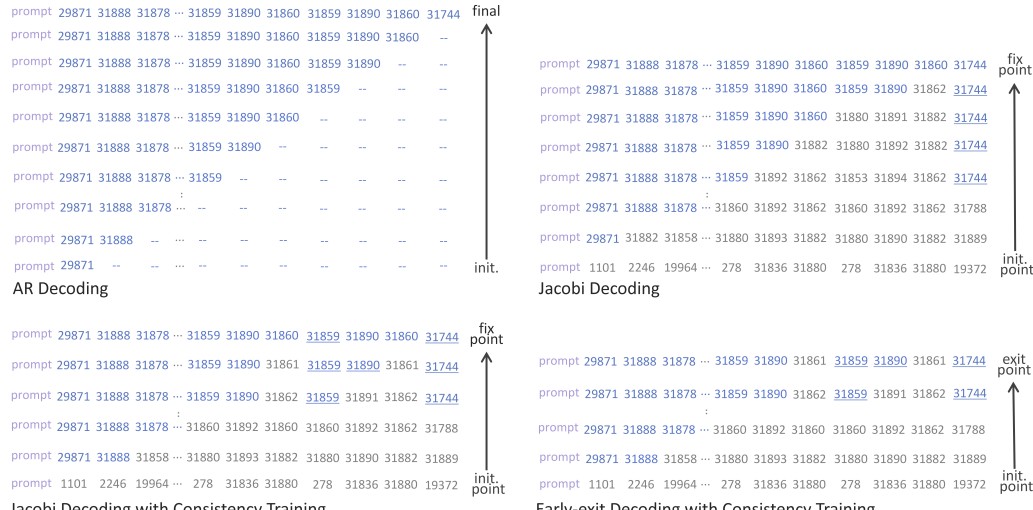

Figure 7: **An instance of Jacobi trajectory with different decoding methods.** Gray numbers indicate incorrect tokens, while blue numbers denote correct ones. Blue numbers with underlines represent fixed tokens.

## F VISUALIZATION OF JACOBI TRAJECTORY WITH EARLY-EXIT DECODING.

We randomly sample an image-text pair and visualize the decoding trajectory of different methods in Figure 7. AR decoding predicts all 37 tokens sequentially, requiring 37 forward passes in total. Therefore, it results in the longest decoding trajectory and introduces considerable computational redundancy. Jacobi decoding starts from a randomly initialized state and updates all tokens simultaneously in each forward pass until convergence to a fixed point. It requires fewer iterations to reach the fixed point, thereby accelerating the decoding process. After consistency training, the model's ability to generate accurate parallel predictions during Jacobi decoding is significantly improved. As a result, even fewer iterations are needed to achieve convergence, further enhancing decoding efficiency. Early-exit decoding enables our method to terminate at an intermediate exit point, which is typically reached earlier than the fixed point. This relaxes the strict convergence requirement during decoding, leading to the fewest forward passes and the highest decoding efficiency. Then we analyze the trajectory of each decoding methods in detail:

**AR Decoding**. As shown in the top-left corner of Figure 7, AR decoding is the conventional and widely adopted approach in sequence generation tasks. It generates tokens sequentially in a left-to-right manner, where each token depends strictly on all previously generated tokens. This method ensures stable performance and high output accuracy. However, its inherently sequential nature prevents parallelization, leading to slow inference and becoming a major bottleneck in real-time applications.

**Jacobi Decoding**. As shown in the top-right corner of Figure 7, Jacobi decoding draws inspiration from the classical Jacobi iterative method in numerical computation. It enables simultaneous updates of all tokens by iteratively refining predictions based on the outputs from the previous step. While this method successfully breaks the sequential dependency of autoregressive decoding and supports parallelism, its convergence speed remains relatively slow. Without specialized training strategies, its acceleration potential is limited in practice.

**Jacobi Decoding with consistency training** . As shown in the Bottom-left corner of Figure 7, Consistency training is introduced to enhance the model's ability to converge from arbitrary initial predictions toward a fixed point. By encouraging the prediction of more accurate tokens across iterations, it significantly reduces the number of steps required for convergence, thereby improving decoding efficiency. Despite this advancement, inference still suffers from inefficiency points, where certain tokens require more iterations to stabilize. These outlier tokens constrain the minimum achievable decoding latency, leaving room for further optimization.

**Early-exit Decoding with consistency training** . As shown in the Bottom-right corner of Figure 7, building on consistency training, the early-exit decoding with the consistency training strategy allows token-wise termination once predictions converge or exhibit negligible change. Alternatively, a global iteration cap can be imposed to force early termination across all tokens. The empirical results shown in Figure 3, demonstrate that models trained with consistency loss maintain robust performance even under strict iteration limits. Notably, limiting the number of iterations at certain decoding steps does not significantly affect the overall task completion rate.

By striking an elegant balance between speed and performance, early-exit Jacobi decoding represents one of the most practical and efficient solutions for parallel inference.

## G  FUTURE WORK

Recent works investigate more efficient tokenization schemes for robot actions in autoregressive VLAs (Pertsch et al., 2025; Belkhale & Sadigh, 2024). They allow representing $x_2$ action components with $x_1$ tokens where $x_1 < x_2$, thereby improving the efficiency of model training and inference, as well as the capture of high-frequency actions. Combining our method with these works to further achieve real-time inference is a promising direction. Our method also demonstrates potential to drive progress across embodied chain-of-thought (Zawalski et al., 2025) that produce extensive intermediate reasoning tokens and knowledge insulating (Driess et al., 2025) that require simultaneous generation of both linguistic responses and action tokens.

## H  COMPARED METHODS

**FastV (Chen et al., 2024).** FastV is a plug-and-play inference-side module for vision–language models that reduces excessive attention on visual tokens in deep layers. It learns adaptive sparsity in early blocks and prunes low-contribution visual tokens later, cutting compute and latency without retraining or architectural changes. On image understanding benchmarks (e.g., A-OKVQA (Schwenk et al., 2022)), a representative setting (K=2 and R=50%) yields about 45% FLOPs reduction with essentially unchanged accuracy, and the speed–accuracy trade-off is tunable by decreasing K and increasing R. Latency measurements show that a 13B model with FastV can run at 7B-like speed while outperforming it on A-OKVQA. On video question answering typically saves more than 40% computation and can improve accuracy, particularly on TGIF (Jang et al., 2017), consistent with higher token redundancy in video (about 576 tokens per image in LLaVA-1.5 (Liu et al., 2024b) and about 2048 per clip in Video-LLaVA (Lin et al., 2024a)). Overall, FastV provides sizable efficiency gains at negligible accuracy cost and can be tuned to different deployment budgets.

**SparseVLM (Zhang et al., 2025c).** SparseVLM is a text-guided, training-free token optimization method that prunes visual tokens without introducing extra parameters or fine-tuning costs. The method uses self-attention with text tokens to score visual-token relevance, applies a rank-based strategy to adaptively set per-layer sparsity, and recycles pruned tokens into compact representations to mitigate information loss. In practice, pruning from 576 to 192 tokens yields an average accuracy drop of about 0.9% and outperforms ToMe (Bolya et al., 2023) by 10.2%. With only 64 tokens retained, it surpasses FastV by 17.3%. On video question answering, it typically saves over 40% computation and reaches 95.0% average accuracy under the same 194-token budget, compared to 80.3% for FastV.

**PD-VLA (Song et al., 2025).** PD-VLA tackles the decoding bottleneck of autoregressive VLAs with action chunking by recasting sequential decoding as a parallel fixed-point problem solved over a small number of iterations. All action tokens within a chunk are updated simultaneously, which enables training-free acceleration and preserves compatibility with existing speedups while keeping policy quality largely intact. In head-to-head comparisons with representative policies, PD-VLA delivers competitive manipulation performance and shows clear gains over the fundamental LLaVA-VLA baseline, supporting its effectiveness as a decoding upgrade. An ablation on decoding horizon shows that updating longer sequences in parallel improves both modeling fidelity and throughput. The horizon of 37 achieves the strongest manipulation ability and the highest decoding speed, while the horizon of 7 outperforms 16 as it better matches the distribution of the single action. As the horizon grows, the number of fixed tokens increases and decoding speed rises from 41.48 to 52.84 tokens per second, though redundant tokens at 16 can depress execution frequency.

The speed distribution confirms this trend. At horizon 37 the maximum speed is roughly twice that of horizon 7 and of autoregressive decoding due to fewer required iterations. Across simulation and real-world tests, PD-VLA attains higher execution frequency with competitive success rates. On a 7-DoF manipulator it reaches 2.52× control frequency.

