# OpenReview forum: "CEED-VLA: Consistency Vision-language-action Model with Early-exit Decoding"
_ICLR.cc/2026/Conference — ICLR 2026 Conference Withdrawn Submission_

### Official Review · Reviewer_CiWo · 2025-10-21

**Soundness:** 2
**Presentation:** 2
**Contribution:** 2
**Rating:** 2
**Confidence:** 4

**Summary:**

The paper proposes CEED-VLA, a method for speeding up VLAs by applying a consistency distillation framework inspired by numerical fixed-point solvers. The idea is to train a student VLA to perform “Jacobi decoding,” where instead of generating actions autoregressively (token by token), it predicts all tokens in parallel and iteratively refines them toward a consistent solution. The authors collect these “Jacobi trajectories” from a pretrained autoregressive teacher and use them to train the student with a consistency loss, mixing teacher-generated and ground-truth supervision to stabilize learning. The goal is to achieve faster inference without losing much accuracy, and the paper reports speedups of around 2–3× on manipulation and simulation benchmarks.

**Strengths:**

1. The paper applies consistency learning to VLAs, making them predict robot actions faster without changing the overall design.

2. It directly addresses the slow inference problem of VLAs and shows up to four times faster performance with minimal accuracy loss.

3. The experiments are thorough, including both simulated and real robot tests.

**Weaknesses:**

1. The central limitation of this paper lies in how it positions CEED-VLA within the broader landscape of non-autoregressive consistency-based policy generation, especially recent approaches that use diffusion or flow-matching architectures, such as Pi0.5, ManiCM, or FlowPolicy. These works have already demonstrated that consistency-style distillation can accelerate robot action inference by learning direct mappings from noisy or intermediate latent states to final actions. For example, FlowPolicy uses consistency flow matching to generate 3D action trajectories in a single step, and ManiCM applies consistency constraints to diffusion models to achieve near real-time inference. The paper even cites these efforts but stops short of explaining how its Jacobi-based consistency mechanism offers a practical or theoretical advantage over those diffusion-based counterparts .In particular, CEED-VLA’s proposed “Jacobi trajectory consistency” could be interpreted as a reformulation of the same fixed-point mapping ideas already used in diffusion consistency models, except applied to token sequences instead of continuous noise states. However, the paper doesn’t provide any cross-framework comparison, either empirically or conceptually. There’s no discussion of whether CEED-VLA achieves comparable *speed-quality tradeoffs* to diffusion or flow-matching policies, or if its discrete token formulation introduces new advantages (e.g., better multimodal grounding) or limitations (e.g., reduced smoothness of the action space).

    The absence of such comparisons leaves the me unsure how significant this work really is beyond its immediate architectural niche. It’s unclear whether CEED-VLA meaningfully advances beyond prior diffusion-based acceleration methods or simply adapts similar ideas to a VLA-specific setting.

2. One of the main weak points in this paper is how it handles the teacher model during the so-called “Jacobi decoding” phase. The authors repeatedly state that AR-VLAs struggle to perform Jacobi decoding because each token depends on the correctness of previous ones. In other words, if an early token is wrong, the whole sequence collapses, which makes sense for AR training. But then, right after making that argument, the paper says it “prompts the teacher VLA to perform Jacobi decoding” to collect trajectories for consistency training. That’s a bit contradictory. If the teacher model can’t meaningfully perform Jacobi decoding, how can it be relied on to produce high-quality Jacobi trajectories for the student? The paper doesn’t really explain how this works in practice. On top of that, the method for generating these trajectories isn’t clearly described. There’s no detail about how many iterations are run, how the intermediate steps are stabilized, or how the authors ensure convergence. So, it’s hard to tell whether these “Jacobi trajectories” are genuine iterative refinements or just a few repeated AR predictions packaged as Jacobi decoding. Even if we assume the teacher can produce some kind of iterative data, the paper doesn’t evaluate how reliable or noisy those trajectories are. The authors use “mixed-label supervision” to fall back on ground truth when the teacher is wrong, but they never quantify how often that happens, which makes it hard to judge how much the distillation actually depends on the teacher’s outputs.

    Overall, this part of the paper feels underdefined. It introduces a conceptual inconsistency (using a model that supposedly can’t do Jacobi decoding to generate Jacobi data) and leaves out key implementation details that would be necessary to reproduce the results.

**Questions:**

1.	CEED-VLA’s idea of Jacobi trajectory consistency feels conceptually close to the consistency principles used in diffusion and flow-matching models. Could you clarify what new insight this discrete, token-based framing adds beyond those earlier formulations?
2.	More broadly, how should we think about CEED-VLA’s contribution? is it a general consistency-learning framework that happens to apply to VLAs, or mainly a VLA adaptation of existing diffusion-style ideas?
3. Does the Jacobi-based consistency mechanism actually change what the model learns about action representation, or is its benefit primarily about faster inference?

---

### Official Review · Reviewer_fNEC · 2025-10-23

**Soundness:** 3
**Presentation:** 3
**Contribution:** 3
**Rating:** 6
**Confidence:** 4

**Summary:**

This submission aims to accelerate Jacobi decoding in VLAs through two perspectives: 1) training: a trajectory from a teacher model is collected and used to distill; 2) inference: an early exiting strategy is proposed. The proposed method achieves significant speedup.

**Strengths:**

1. The motivation is clear, and the idea is straightforward and reasonable;

2. The experiments show good speedup;

3. The ablation study is clear, showing the effects of different proposed components.

**Weaknesses:**

1. A question about the early exiting strategy: it seems that an exiting point $\sigma$ is predetermined and fixed before inference, and this exiting point decides a trade-off between performance and efficiency.
    - From my viewpoint, early exiting could be realized as a data-dependent strategy, which has been extensively studied in dynamic neural networks [1], including the fields of computer vision [2,3], LLM [4], or VLA [5].
    - Therefore, I wonder about this design choice of the authors: will a data-dependent (such as confidence-based) strategy further improve the trade-off?

2. I notice that Deer-VLA [5] is mentioned in the related works. Since Deer-VLA is a representative early-exiting method for efficient VLA, it is recommended to conduct an empirical comparison with it.

3. Another suggestion about the compared baselines: the proposed method can be viewed as  distillation training + early exiting, and it is compared with AR or Jacobi decoding without distillation training or early exiting. It will be more convincing if the method is compared with a baseline with early exiting.

[1] Dynamic Neural Networks: A Survey. TPAMI, 2021.

[2] Resolution adaptive networks for efficient inference. CVPR, 2020.

[3] Dynamic perceiver for efficient visual recognition. ICCV, 2023.

[4] Kangaroo: Lossless self-speculative decoding for accelerating llms via double early exiting. NeurIPS, 2024.

[5] Deer-vla: Dynamic inference of multimodal large language models for efficient robot execution. NeurIPS, 2024.

**Questions:**

I'm also curious about the relationship between the proposed method and:

1.  speculative decoding, which performs early exiting;

2. the recent popular diffusion LLM [6], which performs a similar parallel "rewriting" process for language tokens;

3. multi-token prediction [7].

[6] Large Language Diffusion Models

[7] DeepSeek-V3 Technical Report

---

### Official Review · Reviewer_yJwS · 2025-10-26

**Soundness:** 3
**Presentation:** 3
**Contribution:** 2
**Rating:** 4
**Confidence:** 4

**Summary:**

The paper tackles the inference-speed bottleneck of Vision-Language-Action (VLA) models—crucial for high-frequency, dexterous robot control.
It builds upon Jacobi (parallel) decoding, which reformulates autoregressive (AR) token prediction as a fixed-point iteration, and using early exit to avoid the whole decoding steps which including many unnecessary steps.

**Strengths:**

1. The motivation to accelerate RT-2 style VLA is very reasonable. The motivation is quiet clear.
2. Lots of ablations and design choice is verified during experiments.

**Weaknesses:**

1. Although the motivations of this paper are reasonable, the techniques employed are largely based on existing methods. The Jacobi decoding strategy was originally proposed for language models (https://arxiv.org/pdf/2403.00835), and many implementation details appear to be directly borrowed from that work. The primary contribution of this paper, therefore, lies in adapting an existing LLM acceleration method to the VLA setting.

2. Several existing works have explored alternative approaches to accelerate VLAs by leveraging the unique properties of robotics tasks. For example, the Knowledge Insulating paper (https://arxiv.org/abs/2505.23705) first trains a VLA with an autoregressive loss, then freezes the VLM backbone and introduces an action expert to accelerate action generation. This design achieves high control frequency while maintaining the generalization ability of the VLM.

**Questions:**

1. Does the author observe any additional benefit when apply Jacobi decoding to VLA models? Is the method able to leverage some unique structural properties of robotics tasks? Otherwise, it appears to be merely an application of an existing LLM method to VLAs.

---

### Official Review · Reviewer_Kkay · 2025-11-02

**Soundness:** 1
**Presentation:** 2
**Contribution:** 1
**Rating:** 2
**Confidence:** 4

**Summary:**

This paper proposes CEED-VLA, an acceleration framework for Vision-Language-Action (VLA) models based on Jacobi decoding with consistency distillation. The key idea is to reinterpret autoregressive (AR) decoding as a fixed-point iteration problem and to train a student model that can directly map intermediate Jacobi states to the converged action sequence. With the addition of an early-exit criterion, the approach achieves 2×–4× inference speedup over standard AR decoding on CALVIN and LIBERO benchmarks.

**Strengths:**

The acceleration results are quantitatively significant and consistently shown on two base models (LLaVA-VLA and OpenVLA).

**Weaknesses:**

- Lack of discussion on the necessity of autoregression.
The paper’s motivation assumes the continued importance of AR decoding, but this assumption is not well-justified in the introduction.
Since many modern heads (bi-directional attention, diffusion, or flow matching) are inherently parallel, the authors should explicitly discuss when and why AR decoding remains necessary for manipulation models.

- Choice of baselines and comparison.
The experiments are conducted on LLaVA-VLA and a chunk-based OpenVLA, but neither is a widely-adopted standard baseline.
  -  LLaVA-VLA itself originates from a parallel decoding paper (PD-VLA) rather than a canonical VLA benchmark. π₀-FAST is a better baseline to validate this idea.
  - OpenVLA is artificially modified with action chunking, while the community now has OpenVLA-OFT, which already supports fully parallel (one-iteration) decoding. Comparing CEED-VLA with OpenVLA-OFT should provide a stronger acceleration baseline—would be more convincing.

- Control assumption and flexibility.
The reported acceleration assumes no receding-horizon control, i.e., the model predicts an entire action chunk and executes it fully.
This setup is not widely used in real-world control, as it reduces adaptability to dynamic changes. If the target scenario is static manipulation, please expicitely explain about it. Moreover, if in static scene case, one could achieve similar speed-ups by trajectory down-sampling, which is conceptually simpler. The authors should clarify why their acceleration scheme is preferable under such assumptions.

**Questions:**

- How does the proposed Jacobi + Distillation framework differ in performance from diffusion or consistency models?
- Under what conditions does maintaining an AR formulation remain beneficial or necessary in VLA research?

---

### Official Review · Reviewer_V83y · 2025-11-02

**Soundness:** 3
**Presentation:** 3
**Contribution:** 3
**Rating:** 4
**Confidence:** 4

**Summary:**

**Problem.** Vision–Language–Action (VLA) models suffer from slow inference, limiting high-frequency/dexterous manipulation. **Method.** The paper reframes decoding as Jacobi fixed-point iterations and proposes (i) **consistency distillation** so a student can map any intermediate Jacobi state to the fixed point, plus (ii) **mixed-label AR supervision** to curb teacher-error accumulation, and (iii) **early-exit decoding** that relaxes strict convergence. **Key innovations.** A KL-style consistency loss that conditions student predictions on intermediate vs fixed-point states (Eq. (3)), mixed-label switching by an L1 threshold δmax (Alg. 1), and a fixed exit point σ to cap iterations. **Main results.** On CALVIN with LLaVA-VLA, CEED-VLA doubles tokens/s and 3.3× execution frequency with negligible performance change; on LIBERO-Long with OpenVLA, it reaches 4.1× speedup and 4.3× frequency; real-robot tasks show large gains, especially for dexterous control (e.g., towel fold 75% vs 5% baseline).

**Strengths:**

- **Well-scoped training recipe.** Consistency loss + mixed AR loss with an explicit switch by δmax; concise algorithmic summary (Alg. 1).
- **Early-exit decoding matters in practice.** Identifies long-tail “redundant iterations”; early-exit improves **min** and **avg** speed (Table 1).
- **Compelling empirical gains.** 2.0× on CALVIN (tokens/s 79.2 vs 39.6) and 4.1× on LIBERO-Long; frequency up to 25.6 Hz with OpenVLA (Table 2).

**Weaknesses:**

1) **Loss formulation clarity.**
Eq. (3) appears to optimize a KL between *Q*θ−(· | Y\*_i, s, l) and *Q*θ(· | Y_i, s, l), i.e., the numerator is conditioned on the **fixed-point token** Y\*_i while the denominator is conditioned on the **intermediate token** Y_i. Please supply a derivation that this realizes “map any Y on J to Y\*” rather than merely aligning two different conditionals; include why the stop-gradient teacher is the same student and whether this collapses to self-distillation with distribution shift across conditionings. Also clarify per-token independence assumptions in the summation over *i*. (Eq. (3), §3.3.1).

2) **Mixed-label thresholding details.**
Algorithm 1 gates labels using an L1 threshold δmax, but the paper does not specify how δmax is set/tuned nor its robustness across datasets (only a qualitative motivation in Fig. 6). Please report a sensitivity curve of δmax vs speed/length/success and how often teacher labels are replaced, stratified by task. (Alg. 1; Appendix D Fig. 6).

3) **Early-exit criterion is fixed and non-adaptive.**
The current exit point σ is a global cap; ablations (Fig. 3) sweep σ, but an adaptive exit (e.g., token-wise stability, small Δ logits, or confidence thresholds) might further cut tail latency without accuracy loss. Please compare fixed-σ vs adaptive criteria and report worst-case latency (p95/p99) and failure modes on long-horizon subtasks. (Table 1; Fig. 3).

4) **Speed reporting and fairness.**
Table 2 mixes devices across base models (4090 for CALVIN, H100 for LIBERO-Long). While relative “×” speedups are per-setting, absolute tokens/s and Hz can be device-dependent. Please: (i) confirm identical hardware across compared methods **within each setting**, (ii) report wall-clock latency per action including pre/post-processing, and (iii) add p95/p99 latency to capture tail behavior that matters for control. (Table 2 caption).

5) **Novelty positioning vs prior parallel decoding.**
PD-VLA already reformulates AR as parallel fixed-point updates; your novelty is the **consistency distillation** + **mixed-label AR** + **early-exit**. Please make this contrast explicit with a side-by-side training/runtime schematic and a controlled study where PD-VLA is augmented with each of your components to isolate the source of gains. (Related Work §2.1; Table 2).

6) **Theoretical guarantees.**
The intro claims Jacobi converges to greedy AR outputs (fixed point), but the conditions ensuring convergence under your modified training (consistency + early-exit) are not analyzed. Please provide assumptions (e.g., contraction around Y\*, stability bounds) or empirical fixed-point accuracy vs AR-greedy equivalence across datasets. (Intro/§3.1; Fig. 7 narrative).

7) **Real-world evaluation breadth and statistics.**
Real-robot results cover 4 tasks ×20 rollouts; large gains are promising (avg 77.5% vs 33.25%), but please add confidence intervals, per-task significance tests, and safety/failure analyses (e.g., early-exit-induced instabilities). Also report runtime breakdowns (perception vs policy vs actuation) and any controller smoothing. (Table 6; Fig. 4).

8) **Data and compute cost of trajectory collection.**
Jacobi-trajectory collection (Alg. 2) can be expensive. Provide the total wall-clock cost, failure rate before convergence, and storage footprint for 60k–120k trajectories; include an ablation on reduced trajectories combined with stronger mixed-labeling. (Alg. 2; Fig. 3 right).

9) **Reproducibility.**
Please release code, scripts for Jacobi prompts, teacher checkpoints, δmax/ω settings, and exact σ used per experiment; the current text gives high-level recipes but not config tables. (Sections 3–4).

**Questions:**

See weaknesses

---

### Note · Authors · 2025-11-14

**Comment:**

Thanks for the comments of the reviewers.

**Withdrawal Confirmation:**

I have read and agree with the venue's withdrawal policy on behalf of myself and my co-authors.